# A microbial safari: finding evidence of *Mycobacterium bovis* DNA in soil from the Kruger National Park, South Africa

M. C. Matthews,[1] M. E. M. Toorians,[2,3] T. J. Davies,[3,4] R. D. Stewart,[5] W. J. Goosen,[1,6] M. A. Miller[1]

**ABSTRACT** The primary pathogen causing animal tuberculosis (TB), *Mycobacterium bovis*, can infect a wide range of mammals, including humans. This pathogen is considered endemic in the Kruger National Park (KNP), South Africa, where it threatens the health and conservation of African wildlife, including endangered species. This mycobacterial pathogen is spread through close contact with an infected host by aerosols or ingestion but may also be transmitted indirectly through the environment. Detecting environmental *M. bovis* is challenging due to the complexity of the sample matrix and may require culture-independent techniques due to biosecurity restrictions associated with sample movement. In this study, DNA was extracted from soil near water sources in the KNP ($n = 180$) and screened for *Mycobacterium tuberculosis* complex (MTBC) DNA using the *hsp65* polymerase chain reaction (PCR) followed by Sanger amplicon sequencing and the GeneXpert MTB/RIF Ultra (GXU) qPCR assay. The region of difference (RD)-PCR was used to confirm whether *M. bovis* DNA was present. Sanger *hsp65* amplicon sequencing and GXU detected MTBC DNA in three and zero samples, respectively. Moreover, Sanger sequencing detected organisms belonging to the *Mycobacterium* genus in 44 samples and non-tuberculous *Mycobacterium* species in 21 samples. The RD-PCR confirmed *M. bovis* DNA presence in 2/3 MTBC-positive samples. The presence of *M. bovis* DNA in KNP soil suggests potential environmental contamination by shedding from infected animals. Further research is required to confirm the viability of MTBC and the role of environmental contamination in the TB epidemiology of this multi-host system.

**IMPORTANCE** This article describes the first evidence that DNA from the bacterial pathogen, *Mycobacterium bovis*, which causes animal tuberculosis (TB) in wildlife, can be detected in soil from the Kruger National Park (KNP). Animal TB threatens wildlife conservation, including threatened and endangered species, in areas such as the KNP. Pathogenic *Mycobacterium* are spread primarily through direct contact with infected hosts. However, the presence of *M. bovis* DNA in KNP soil could indicate a role for the environment in disease transmission. This complements the growing evidence from European regions that *M. bovis* can be shed by infected animals into water, soil, or plant material and potentially infect animals in the surrounding environment. This indirect route of spread has implications for disease management strategies and warrants further scientific investigation. Moreover, the direct DNA-based detection techniques described in this study may provide a tool for detecting *Mycobacterium* pathogens using non-invasive sampling (sampling the environment rather than animals directly) when culturing is not possible.

**KEYWORDS** animal tuberculosis, culture independent detection, environmental *Mycobacterium*, GeneXpert MTB/RIF Ultra, *hsp65* PCR, indirect transmission, *Mycobacterium bovis*, *Mycobacterium tuberculosis* complex, Sanger sequencing

**Peer Reviewers** Jayne Hope, Roslin Institute, Easter Bush, Midlothian, United Kingdom; Eman Khalifa, Matrouh University, Matrouh, Egypt

Address correspondence to M. C. Matthews, megandutoit@sun.ac.za.

W. J. Goosen and M. A. Miller contributed equally to this article.

The authors declare no conflict of interest.

See the funding table on p. 10.

In South Africa, *Mycobacterium tuberculosis* complex (MTBC) infections pose significant risks to human and animal health (1, 2). The human-adapted MTBC, *Mycobacterium tuberculosis* (MTB), causes tuberculosis (TB), a chronic disease of global human health concern (3). Likewise, the animal-adapted MTBC ecotype, *Mycobacterium bovis*, causes animal TB, a chronic disease affecting a range of mammalian species, including zoonotic TB in humans (4, 5).

In the Kruger National Park (KNP), South Africa, *M. bovis* is endemic with African buffalo (*Syncerus caffer*) serving as the primary maintenance hosts (6). Currently, *M. bovis* infection has been reported in 16 wildlife species within KNP, including threatened and endangered African rhinoceros (*Diceros bicornis* and *Ceratotherium simum*) and African wild dogs (*Lycaon pictus*), highlighting the potential threat to conservation efforts (7, 8).

Animals infected with *M. bovis* may shed intermittently viable bacilli in secretions, such as aerosols, urine, and feces (9). These organisms can survive in the environment for up to 5 months (10–12). Favorable conditions for environmental persistence include limited UV exposure, moist conditions, and cool temperatures (13–15). Experimental and epidemiological studies, relying on microbiological and molecular detection assays, have shown that viable *M. bovis* can be spread through contaminated environmental material, e.g. soil, water, feed, and pastures (9, 16). The presence of *M. bovis* in oronasal swabs from South African wildlife (17–19) indicates that shedding could result in environmental contamination. A study by Tanner and Michel (10) found that *M. bovis* can persist in tissue samples in the KNP environment; however, Michel et al. (20) were unable to detect *M. bovis* in water troughs used by infected African buffalo.

Methods for detecting environmental MTBC have conventionally relied on mycobacterial culture (21); however, this approach has some limitations. Environmental samples are often paucibacillary, and detection with microbial culture can be confounded by contamination with competing microorganisms and loss of viable MTBC due to harsh decontamination protocols (22, 23). Moreover, mycobacterial cultures require specialized laboratories, lengthy incubation times, and increased biosecurity (24). In South Africa, animal and environmental sample transport to laboratories has been restricted due to the presence of controlled diseases in some areas (25).

Advances in molecular techniques, such as polymerase chain reaction (PCR) and sequencing technologies, have been used to detect MTBC DNA in samples without the requirement for mycobacterial culture (26–28). Since the transport of untreated/unsterilized soil from KNP for mycobacterial culture is restricted, culture-independent methods for *M. bovis* detection are required to determine whether environmental contamination is present. In this study, we investigate whether it is possible to detect *M. bovis* using DNA extracted directly from KNP soil samples. This supports further investigation of the source and role of the environment in animal TB epidemiology in KNP.

## MATERIALS AND METHODS

### Soil sample collection

Soil samples (10 g) were collected from six locations near water sources within KNP, South Africa. The global positioning system coordinates (latitude and longitude) were recorded for each site (Table S1). Briefly, each location was marked 1 m from the water's edge, and samples were collected from 10 randomly selected 0.1 m × 0.1 m sub-squares within a 1 m × 1 m quadrant to avoid selection bias. Five replicate samples were collected per site from topsoil (3 cm in depth), equally spaced along the waterhole circumference. Sampling occurred at six time points in June 2022, and soil samples (*n* = 180) were stored in Ziplock bags at −80°C until processed. Prior to DNA extraction, soil samples were thawed at room temperature overnight (~25°C for 8 h). Samples were transferred to paper bags and placed in an oven at 50°C for 5 h to dry. After drying, soil samples were mixed using a sterile spoon.

## Environmental DNA extraction

Soil DNA was extracted from 0.25 g of each sample using the PowerSoil Pro DNeasy kits (Qiagen, Hilden, Germany), according to the manufacturer's guidelines. A final volume of 50 µL genomic DNA was eluted per sample. Eluted DNA was centrifuged at 5,000 rcf for 5 min, then placed in an oven at 50°C for 2–3 h or until dried. Dry storage of DNA was used as it is comparable to freezing DNA at −80°C (29). In addition, DNA was extracted from 200 µL of *M. bovis* culture isolate as a positive control and 200 µL nuclease-free water (negative control) using the method described above. The *M. bovis* isolate was grown in a Mycobacterial Growth Indicator Tube (Becton Dickinson, NJ, USA) from a single colony under biosafety level 3 (BSL3) conditions at Stellenbosch University's Biomedical Research Institute (Tygerberg, South Africa). A 1 mL culture aliquot was boiled (100°C for 30 min) to inactivate the sample prior to removing it from the BSL3. The sample was then centrifuged in a Prism microcentrifuge (Labnet, Edison, NJ, USA) at 5,000 rcf for 15 min, and 800 µL of supernatant was discarded prior to DNA extraction.

## Determination of extracted soil DNA quality and quantity

Prior to downstream analysis, dried DNA was reconstituted in 100 µL nuclease-free water. Eighteen DNA samples were selected using a random number generator (30) for quality and quantity testing. Briefly, the mean concentration (ng/µL) of extracted DNA was determined using the Qubit DNA Broad Range Assay Kit and the Qubit 4 Fluorometer (Thermo Fisher Scientific, Waltham, MA, USA) according to the manufacturer's instructions. The presence and integrity of the DNA was confirmed by gel electrophoresis using a 1% agarose gel and visualization with the Bio-Rad ChemiDoc Universal Hood III and Gel Documentation System (Bio-Rad Laboratories, Hercules, CA, USA). A 5 µL aliquot of DNA was added to each well, and a 100 bp Plus GeneRuler (Thermo Fisher Scientific) was used to estimate the size of extracted DNA. The DNA samples were also diluted 1:10 and amplified in a 16S PCR, as per Matthews et al. (31), to ensure variation in soil composition did not result in bias or co-extraction of inhibitors and downstream PCR inhibition.

## Screening soil for *Mycobacterium* or MTBC DNA based on PCR and Sanger amplicon sequencing (SAS)

### Amplification of Mycobacterium DNA with a genus-specific PCR

Diluted DNA (1:10) from all 180 KNP soil samples was used in a *Mycobacterium* genus-specific *hsp65* PCR (32). Briefly, OneTaq Hot Start 2× Master Mix with Standard Buffer (New England Biolabs,Ipswich, MA, USA) was used according to manufacturer's guidelines. Each reaction contained 0.5 µL of forward primer (10 µM), 0.5 µL of reverse primer (10 µM), and 2 µL of DNA template, using primer sequences previously described (31). A non-template control (nuclease-free water) and both positive and negative controls (described above) were included in each PCR to validate assay performance and rule out cross-contamination. A dilution series of *M. bovis*-positive control DNA ($10–10^{-5}$ ng/µL) in nuclease-free water was used to evaluate the sensitivity of the *hsp65* PCRs. Cycling conditions consisted of one cycle at 94°C for 1 min, followed by 35 cycles of denaturation at 94°C for 30 s, annealing at 62.5°C for 30 s, elongation at 68°C for 90 s, and a final elongation step of 5 min at 72°C, using an Applied Biosystems Veriti 96-Well Fast Thermal Cycler (Thermo Fisher Scientific). The presence and size of PCR amplicons were confirmed by gel electrophoresis as described above.

### Speciation of Mycobacterium based on Sanger amplicon sequencing

Amplicons generated from KNP soil DNA and all controls were submitted for clean-up and Sanger sequencing at the Stellenbosch University Central Analytical Facility (Stellenbosch, South Africa), as previously described (18). Consensus sequences were prepared using BioEdit Sequence Alignment Editor (version 7.7, Tom Hall, CA, USA) and compared to available sequence databases using the National Center for Biotechnology

Information (NCBI) Basic Local Alignment Search Tool for Nucleotides program (https://blast.ncbi.nlm.nih.gov). The percentage coverage ($P_C$) and identity match ($P_{IM}$) between Sanger consensus sequences and sequences on the NCBI database were recorded. Sequences were considered unassigned if $P_C$ was <90% and if $P_{IM}$ was <80%. Sequences with $P_C$ >90% and $P_{IM}$ of 80%–89% were identified to the genus level. If $P_{IM}$ was ≥90% with a known *Mycobacterium* species, the sample was assigned to species or species complex level. Since the *hsp65* gene region does not differ significantly within *Mycobacterium* species or complexes, sequences could not be identified to the ecotype/subspecies level based on amplicon sequencing (33). For example, if a sequence matched an *M. avium* subsp. *paratuberculosis hsp65* sequence from the NCBI database, in this study, it would be identified as *M. avium* complex (MAC). Similarly, a consensus sequence matching an *M. tuberculosis hsp65* sequence on NCBI would be identified as MTBC.

## Differentiating MTBC ecotype DNA based on region of difference (RD)-PCR

If MTBC DNA was detected with *hsp65* PCR SAS, an RD-PCR was performed on these samples to identify *M. bovis* (34). This method was modified by using OneTaq Hot Start Master Mix according to the manufacturer's instructions to reduce non-specific amplification (31). Briefly, RD1 and RD4 PCRs were used to amplify sample DNA, positive controls (DNA extracted from *M. bovis* and *M. tuberculosis* cultures), and a non-template control (nuclease-free water). Amplicons were then visualized using gel electrophoresis as described previously. The RD1 and RD4 amplicon band sizes were used to determine if *M. bovis* DNA was present and to differentiate it from *M. bovis* BCG and other MTBC (34). If multiple amplicons of varying sizes were present, *M. bovis* detection could not be confirmed.

## Screening for MTBC DNA using the Cepheid GeneXpert MTB/RIF Ultra (GXU) qPCR assay

Samples identified to contain *Mycobacterium* DNA based on *hsp65* PCR sequencing results were screened with the Cepheid GXU qPCR assay (Cepheid, Sunnydale, CA, USA) to independently determine the presence of MTBC DNA. A 20 µL aliquot of extracted soil DNA was diluted in 980 µL nuclease-free water. An equal volume of sample lysis buffer (1 mL) was added. Samples were vortexed for 10 s before and after a 10 min incubation at room temperature (20°C–22°C). The total volume (2 mL) was loaded into a GXU cartridge and analyzed according to the manufacturer's guidelines (Cepheid). The GXU outputs included ERROR, INVALID, MTB NOT DETECTED, or MTB DETECTED, with an indication of semi-quantitative bacterial DNA levels and rifampicin (RIF) resistance. Any sample with MTB NOT DETECTED was reported as a negative result (i.e., neither *IS6110* and *IS1081* nor *rpoB* were amplified). Semi-quantitative levels of MTB DETECTED were based on preprogrammed *rpoB* cycle threshold (Ct) values: very low (Ct > 28), low (Ct 22–28), medium (Ct 16–22), or high (Ct < 16). If MTB was not detected with *rpoB* probes but was detected with *IS6110* and *IS1081*, RIF resistance could not be determined, and the GXU output was MTB TRACE DETECTED (35). As GXU cannot distinguish between MTBC members, MTB positive results at high to very low levels were considered positive for MTBC DNA. The confidence that MTBC DNA was present in samples with MTB TRACE DETECTED results is lower and requires confirmation with another molecular technique (36). Therefore, the sensitivity of the GXU was evaluated using a dilution series ($10–10^{-5}$ ng/µL) of *M. bovis*-positive DNA in nuclease-free water. Additionally, the aforementioned negative DNA extraction control was tested to ensure cross-contamination did not occur.

## RESULTS

### Evaluating the quality and quantity of DNA extracted from soil

Replicate soil samples ($n = 5$), collected at six sites at six time points around water sources in the southern KNP in June 2022 (Fig. 1), underwent DNA extraction ($n = 180$). Extracted DNA from a subset of 18 soil samples had an average DNA concentration of

122.40 ng/µL (95% confidence interval: 92.74–152.05 ng/µL) (Table S1). Gel electrophoresis showed intact DNA was present and could be amplified in 88% (16/18) of samples, using 16S PCR (data not shown).

## Screening soil for *Mycobacterium* or MTBC DNA based on PCR and Sanger amplicon sequencing

The *Mycobacterium* genus-specific *hsp65* PCR produced amplicons from 137 (76%) of 180 KNP soil DNA samples (Table S2). Neither the non-template nor negative DNA controls produced amplicons. Serial dilutions of extracted DNA from the *M. bovis*-positive control were successfully amplified for concentrations ranging from $10^{-4}$ to 10 ng/µL (Table 1).

Sanger sequencing results identified *Mycobacterium* spp. DNA in 44 out of the 137 samples that generated *hsp65* PCR amplicons. Of these, 24 samples could be speciated, based on $P_C$ and $P_{IM}$ values of ≥98% and ≥90%, respectively. Three soil samples (#32, #71, #75) were determined to contain MTBC DNA (Fig. 1 and 2, Table 2). The remaining 21 samples were identified as non-tuberculous *Mycobacteria*, including MAC, *M. asiaticum, M. chlorophenolicum, M. chubuense, M. komanii, M. kubicae, M. novocastrense, M. psychrotolerans, M. rutiulm,* and *M. tusciae.*

## Differentiating MTBC ecotype DNA based on RD-PCR

Two of the three MTBC-positive soil samples (#71 and #75, both from site 3) were confirmed to contain *M. bovis* DNA, based on RD-PCR (Fig. 1, Table 3). The third sample (#32 from site 1) was designated as undifferentiated MTBC since the RD1 region could not be amplified and the RD4 PCR resulted in multiple amplicons.

## Screening soil for MTBC DNA using the Cepheid GeneXpert MTB/RIF Ultra qPCR assay

Samples, identified to contain *Mycobacterium* DNA based on *hsp65* PCR sequencing results, and the negative control had MTB NOT DETECTED results in the GXU. In contrast, serial dilutions of *M. bovis*-positive control DNA at concentrations of $10–10^{-2}$ ng/µL had MTB detected at medium to low levels, respectively (Table 1). An MTB TRACE result was reported for the positive control *M. bovis* DNA concentration of $10^{-3}$ ng/µL.

## DISCUSSION

Results from this study provide the first evidence of environmental *M. bovis* DNA in KNP, which is an endemically infected ecosystem with multiple known host species (37–39). This is consistent with limited studies of environmental MTBC in African wildlife areas where *M. bovis* is considered endemic (31, 40, 41). The confirmation of *M. bovis* presence in two samples from the same site in the KNP was based on a culture-independent workflow that included *hsp65* PCR of DNA extracted directly from soil, Sanger amplicon sequencing, and RD-PCR. The one MTBC DNA-positive sample, which could not be differentiated based on RD-PCR, likely contained more than one MTBC ecotype. There is >99% genetic homology between MTBC (42); therefore, attributing an ecotype can be difficult in samples where the mycobacteriome is complex (27).

While GXU can provide a rapid, independent PCR-based method for MTBC screening, it failed to detect any MTBC DNA in soil samples in this study; however, low concentrations of the *M. bovis*-positive control were GXU positive. Although developed primarily for MTBC detection from sputum samples, GXU has also been used for screening environmental samples, in which internal controls ensured negative results were not due to PCR inhibition (31, 43, 44). In this study, extracted soil DNA was used, but the negative results may have been due to MTBC DNA levels below the threshold of detection (*M. bovis* $<10^{-3}$ ng/µL). Therefore, GXU might not be a sufficiently sensitive approach to screening paucibacillary samples such as soil. Employing alternative quantitative PCRs for MTBC detection, such as those used (*IS1081, IS6110,* and *Mbp70*) by Didkowska et al. (45), should be considered for screening environmental samples in future studies.

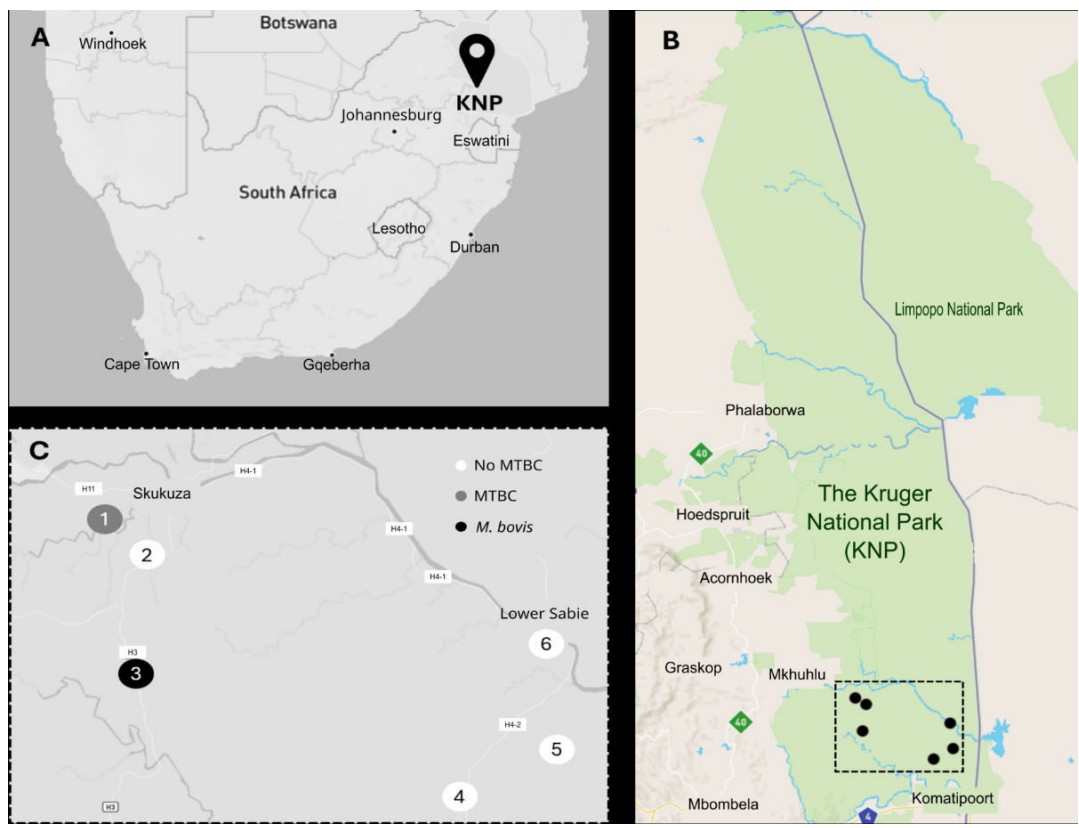

**FIG 1** Soil was collected in South Africa (A) from the Kruger National Park (B) at six locations and screened for DNA from MTBC bacteria and the MTBC ecotype *M. bovis* (C) using PCR and Sanger sequencing.

The low number of MTBC DNA-positive KNP soil samples is notable, since multiple samples were taken near water sources during winter (when surface water is generally scarce in the KNP) to maximize the concentration of wildlife at sampling areas (46). Moreover, African buffalo, the primary maintenance host for *M. bovis* in the KNP, were observed visiting the sampled sites. Research suggests that areas around water sources, where multiple species congregate in high densities, increase the likelihood of detecting environmental MTBC (47, 48). Barasona et al. (47) also showed that mud samples were more likely to be positive for MTBC DNA compared to water. This may explain the lack of *M. bovis* detection in water troughs used by infected KNP buffalo in the study by Michel

**TABLE 1** Serial dilution of cultured *Mycobacterium bovis* DNA detection by Cepheid GXU qPCR assay and *hsp65* PCR and SAS

| Concn[a] | GXU[b] | | Sanger amplicon sequencing[c] | | | |
|---|---|---|---|---|---|---|
| | Result | Ct | PCR | RR | $P_C$ | $P_{IM}$ |
| 10 | MTB medium | 19.0 | Amplified | 3/3 | 100 | 100 |
| 1 | MTB medium | 21.4 | Amplified | 3/3 | 100 | 100 |
| $10^{-1}$ | MTB low | 25.8 | Amplified | 3/3 | 100 | 100 |
| $10^{-2}$ | MTB low | 27.6 | Amplified | 3/3 | 100 | 97 |
| $10^{-3}$ | MTB trace | No Ct | Amplified | 3/3 | 0 | 0 |
| $10^{-4}$ | MTB not detected | No Ct | Amplified | 3/3 | 0 | 0 |
| $10^{-5}$ | MTB not detected | No Ct | NA | 0/3 | 0 | 0 |
| NTC | MTB not detected | No Ct | NA | 0/3 | 0 | 0 |
| Negative | MTB not detected | No Ct | NA | 0/3 | 0 | 0 |

[a]Concentration (Concn) of *Mycobacterium bovis* DNA in nanograms per microliter. No MTBC DNA was present in the non-template control (NTC) or negative control.
[b]The Cepheid GXU qPCR assay was used for semi-quantitative MTBC detection and cycle threshold (Ct) values indicated where applicable.
[c]Extracted DNA was amplified using PCR in triplicate, and the ratio of replicates (RR) that amplified or did not amplify (NA) was indicated. The $P_C$ and $P_{IM}$ between Sanger-sequenced amplicons and MTBC sequences on the NCBI database were also indicated.

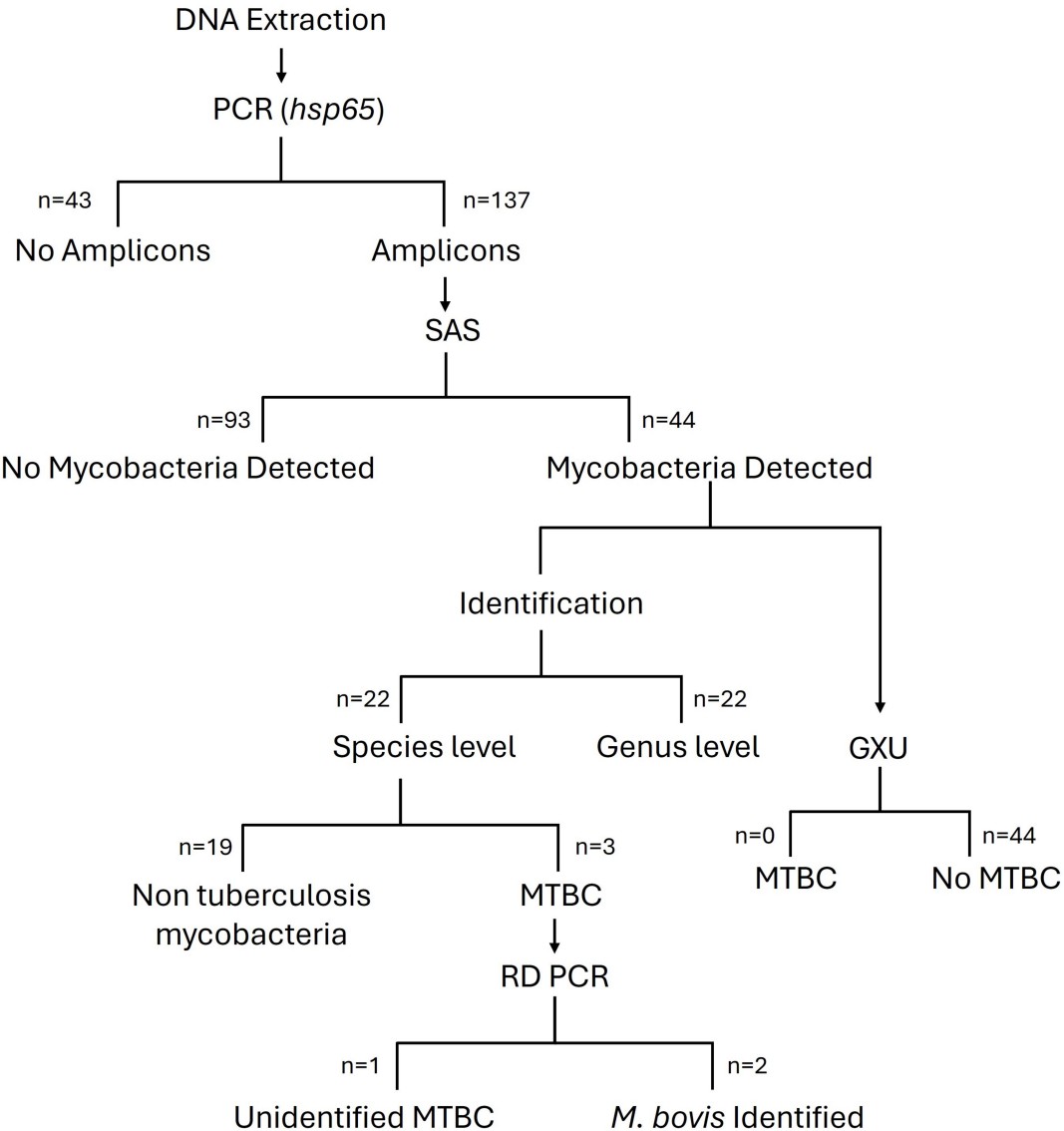

**FIG 2** Flow diagram of screening KNP soil for MTBC and MTBC ecotype *M. bovis* DNA using SAS, GXU qPCR assay, and RD-PCR.

et al. (20). In contrast to studies by Barasona et al. (47) and Matthews et al. (31), soil samples taken around water sources in this study were sandy rather than muddy (pers. obs.). Soil moisture content and mineral composition of the sandy samples may not be optimal for environmental MTBC detection (49).

Another explanation for the low number of positive samples could be poor MTBC persistence under KNP climatic conditions. In a study using experimentally *M. bovis*-spiked soil, bacteria could be detected for 2–12 weeks, depending on abiotic factors (12). Cool, dark, and damp soil conditions support MTBC persistence (11, 12). However, exposure to harsh conditions, such as high temperatures and UV light, can reduce the longevity of bacterial cells or DNA in the environment (10, 12, 13). Therefore, the hot, sunny conditions in KNP (50) may limit *M. bovis* environmental persistence, resulting in fewer positive samples.

The detection of even low numbers of *M. bovis* DNA-positive samples suggests the KNP environment can be contaminated by shedding from infected hosts. Animals infected with *M. bovis* intermittently excrete bacteria in respiratory secretions, urine, and feces, which may contribute to indirect transmission between host species as shown in

**TABLE 2** Soil samples collected near water sources at six different locations in the Kruger National Park, South Africa, in which *Mycobacterium* DNA was detected based on *hsp65* PCR SAS

| No. | Date | Location | PCR | Sanger result | $P_C$ | $P_{IM}$ |
|---|---|---|---|---|---|---|
| 3 | 8/9th | 1 | Amplified | *M. psychrotolerans* | 100 | 90 |
| 4 | 8/9th | 1 | Amplified | *M. psychrotolerans* | 100 | 90 |
| 5 | 8/9th | 1 | Amplified | *M. chlorophenolicum* | 100 | 90 |
| 17 | 8/9th | 4 | Amplified | *M. tusciae* | 100 | 90 |
| 20 | 8/9th | 4 | Amplified | *M. psychrotolerans* | 100 | 90 |
| 32 | 13/14th | 1 | Amplified | MTBC | 100 | 98 |
| 40 | 13/14th | 2 | Amplified | *Mycobacterium* sp. | 100 | 89 |
| 44 | 1/14th | 3 | Amplified | *Mycobacterium* sp. | 100 | 89 |
| 47 | 13/14th | 4 | Amplified | *Mycobacterium* sp. | 100 | 86 |
| 48 | 13/14th | 4 | Amplified | *Mycobacterium* sp. | 100 | 89 |
| 49 | 13/14th | 4 | Amplified | *M. chubuense* | 100 | 91 |
| 56 | 13/14th | 6 | Amplified | *Mycobacterium* sp. | 100 | 87 |
| 57 | 13/14th | 6 | Amplified | *Mycobacterium* sp. | 100 | 91 |
| 61 | 15/16th | 1 | Amplified | *M. psychrotolerans* | 98 | 90 |
| 62 | 15/16th | 1 | Amplified | *M. novocastrense* | 99 | 93 |
| 64 | 15/16th | 1 | Amplified | *Mycobacterium* sp. | 99 | 90 |
| 71 | 15/16th | 3 | Amplified | MTBC | 99 | 99 |
| 75 | 15/16th | 3 | Amplified | MTBC | 100 | 96 |
| 77 | 15/16th | 4 | Amplified | *M. kubicae* | 99 | 90 |
| 78 | 15/16th | 4 | Amplified | MAC | 99 | 90 |
| 79 | 15/16th | 4 | Amplified | MAC | 100 | 90 |
| 83 | 15/16th | 5 | Amplified | *Mycobacterium* sp. | 100 | 88 |
| 86 | 15/16th | 6 | Amplified | *Mycobacterium* sp. | 99 | 82 |
| 107 | 20th/21st | 4 | Amplified | *M. asiaticum* | 100 | 91 |
| 109 | 20th/21st | 4 | Amplified | MAC | 100 | 91 |
| 110 | 20th/21st | 4 | Amplified | *Mycobacterium* sp. | 99 | 90 |
| 113 | 20th/21st | 5 | Amplified | *Mycobacterium* sp. | 100 | 88 |
| 116 | 20th/21st | 6 | Amplified | *Mycobacterium* sp. | 100 | 88 |
| 119 | 20th/21st | 6 | Amplified | *Mycobacterium* sp. | 100 | 86 |
| 122 | 23rd/24th | 1 | Amplified | *M. komanii* | 100 | 90 |
| 123 | 23rd/24th | 1 | Amplified | *M. rutilum* | 100 | 91 |
| 125 | 23rd/24th | 1 | Amplified | *Mycobacterium* sp. | 100 | 86 |
| 137 | 23rd/24th | 4 | Amplified | *Mycobacterium* sp. | 99 | 92 |
| 138 | 23rd/24th | 4 | Amplified | *Mycobacterium* sp. | 100 | 88 |
| 144 | 23rd/24th | 5 | Amplified | *M. psychrotolerans* | 100 | 91 |
| 146 | 23rd/24th | 6 | Amplified | *M. chubuense* | 100 | 92 |
| 151 | 28/29th | 1 | Amplified | *Mycobacterium* sp. | 100 | 88 |
| 152 | 28/29th | 1 | Amplified | *M. novocastrense* | 100 | 90 |
| 153 | 28/29th | 1 | Amplified | *M. chubuense* | 100 | 92 |
| 155 | 28/29th | 1 | Amplified | *Mycobacterium* sp. | 99 | 88 |
| 167 | 28/29th | 4 | Amplified | MAC | 98 | 90 |
| 169 | 28/29th | 4 | Amplified | *Mycobacterium* sp. | 100 | 87 |
| 170 | 28/29th | 4 | Amplified | *Mycobacterium* sp. | 99 | 88 |
| 176 | 28/29th | 6 | Amplified | MAC | 100 | 91 |
| Pos | Not applicable | | Amplified | MTBC | 100 | 100 |
| Neg | Not applicable | | NA | NA | NA | NA |

[a]The sample information includes sample number (1–180), sample location (1–6), and the date of collection in June 2022.
[b]Each soil sample underwent DNA extraction, *hsp65* PCR amplification, and Sanger amplicon sequencing (SAS). Samples which could not be amplified (NA) could not undergo SAS. Sequences with $P_C \geq 90\%$ and $P_{IM} \geq 80\%$ compared to sequences from known *Mycobacterium* were identified to *Mycobacterium* genus level. Moreover, sequences with $P_C$ and $P_{IM} \geq 90\%$ with known *Mycobacterium* species were identified to species or species complex level (e.g., MTBC or MAC).

**TABLE 3** RD-PCR of MTBC DNA extracted from KNP soil samples

| Sample or control | RD-PCR[b] | | |
|---|---|---|---|
| | RD1 | RD4 | Confirmed[c] |
| Samples[a] | | | |
| 32 (site 1) | No amplicons | Absent/present | No |
| 72 (site 2) | Present | Absent | Yes |
| 75 (site 3) | Present | Absent | Yes |
| Controls[d] | | | |
| *M. tuberculosis* | Present | Present | No |
| *M. bovis* | Present | Absent | Yes |
| No DNA | No amplicons | No amplicons | No |

[a]Soil sample number (1–180) and collection site (1–6) where MTBC DNA was detected.
[b]Region of difference (RD1 and RD4) PCRs, performed according to Warren et al. (34), were used to confirm that MTBC detected was *M. bovis*.
[c]If the RD1 region was present (146 bp) and the RD4 region was absent (268 bp), *M. bovis* DNA was confirmed to be present.
[d]Controls included DNA extracted from *M. tuberculosis* and *M. bovis* cultures and a no DNA control containing nuclease-free water.

other ecosystems (9, 15, 16, 47, 51). Recent studies have linked a higher risk of MTBC transmission with pathogen detection in environmental samples in areas of Portugal where infected cattle and wildlife co-exist (27, 52). However, there have been relatively few studies in Africa, where MTBC-infected humans and animals may contribute to environmental contamination (31, 40, 53). The results in this study suggest that further research should investigate the sources of contamination and the role of the environment in TB epidemiology in specific contexts, such as the KNP.

A limitation in this study was that soil could not be processed for mycobacterial culture (due to restrictions on transporting samples without heat inactivation), which is the gold standard for detecting viable MTBC (41). The detection of MTBC DNA does not provide evidence of viability as pathogen DNA can persist in the environment after bacteria are non-viable (12, 13). Advanced molecular techniques have been shown to identify MTBC DNA directly in complex samples with comparable or superior specificity and sensitivity to culture (12, 23, 27, 43). However, detecting viable *M. bovis* in environmental samples is essential to determining potential transmission risk (13, 27, 54).

A further limitation of this study was that MTBC and *M. bovis* DNA was identified based on conserved gene sequences (*hsp65* and the RD) rather than whole-genome sequencing (WGS). Phylogenetic comparison could therefore not be used to compare MTBC or *M. bovis* isolates in this study nor to establish epidemiological links between environmental isolates and isolates from surrounding wildlife populations. For WGS to be used in future studies, enrichment of environmental MTBC with culturing or advanced molecular techniques such as those used by Pereira et al. (27) would be required. In the study by Pereira et al. (27), WGS from environmental *M. bovis* were isolated and found to be phylogenetically similar (<10 single nucleotide polymorphism differences) to those from animal isolates from the same area. These epidemiologically linked contamination events provided evidence of excretion from infected animals into the environment and persistence of viable *M. bovis* in that ecosystem (27).

Despite study limitations, the presence of *M. bovis* DNA in KNP provides initial evidence of environmental contamination in this endemically infected system. These findings are consistent with growing evidence that environmental *M. bovis* may be important in persistence and recurrence of animal tuberculosis in endemic multi-host systems (16, 27). In addition, successful detection of *M. bovis* DNA from small volumes of soil using standard off-the-shelf eDNA extraction kits provides an alternative method to culture for non-invasive environmental detection of pathogenic *Mycobacterium* in endemically infected wildlife populations.

In conclusion, *M. bovis* DNA was detected in soil surrounding KNP water sources for the first time using culture-independent techniques. The most likely source of soilborne

*M. bovis* would be infected wildlife populations, although animals were not sampled in this study. Future studies to investigate the epidemiological link between animals and the environment should include determining the viability of MTBC in soil samples and establishing the phylogenetic relationship between viable *M. bovis* isolates from different sources (animal and environmental) in the surrounding area.

## ACKNOWLEDGMENTS

Financial support for this research was provided by the Wellcome Foundation (grant no. 222941/Z/21/Z), GenPath Africa (grant no. 101103171), which was supported by the Global Health EDCTP3 Joint Undertaking and its members, as well as the Bill & Melinda Gates Foundation, the South African Medical Research Council (SAMRC) Centre for Tuberculosis Research, and the National Research Foundation South African Research Chair Initiative (grant no. 86949). M.E.M.T. and T.J.D. were supported by the NSERC Discovery Grant (RGPIN-2020-04439) awarded to T.J.D. M.E.M.T. was further supported by a Mitacs Globalink Research Grant. The opinions expressed and conclusions arrived at are those of the authors and are not necessarily to be attributed to the funders.

We would like to express our gratitude to the Kruger National Park Scientific Services and their staff for their help setting up and carrying out this project, especially Dr. Danny Govender and Refiloe Mohloding. Additionally, we would also like to thank the African Centre of DNA Barcoding in Johannesburg for their help in storing and processing DNA samples.

## AUTHOR AFFILIATIONS

[1]Division of Molecular Biology and Human Genetics, SAMRC Centre for Tuberculosis Research, Faculty of Medicine and Health Sciences, Stellenbosch University, Cape Town, South Africa
[2]Trinity College Dublin, Dublin, Ireland
[3]Department of Botany, Forest & Conservation Sciences, University of British Columbia, Vancouver, Canada
[4]African Centre for DNA Barcoding, University of Johannesburg, Johannesburg, South Africa
[5]Department of Biological and Agricultural Sciences, Sol Plaatje University, Kimberley, South Africa
[6]Department of Microbiology and Biochemistry, University of the Free State, Bloemfontein, South Africa

## AUTHOR ORCIDs

M. C. Matthews http://orcid.org/0000-0002-8994-2293
M. E. M. Toorians http://orcid.org/0000-0001-6337-8092
T. J. Davies http://orcid.org/0000-0003-3318-5948
R. D. Stewart http://orcid.org/0000-0002-1005-0357
W. J. Goosen http://orcid.org/0000-0001-6614-9084
M. A. Miller http://orcid.org/0000-0002-5883-6076

## FUNDING

| Funder | Grant(s) | Author(s) |
| --- | --- | --- |
| Wellcome Trust | No. 222941/Z/21/Z | W.J. Goosen |
| Bill and Melinda Gates Foundation | No. 101103171 | M.A. Miller |
| National Research Foundation | No. 86949 | M.A. Miller |

## AUTHOR CONTRIBUTIONS

M. C. Matthews, Conceptualization, Data curation, Formal analysis, Investigation, Methodology, Project administration, Visualization, Writing – original draft, Writing – review and editing | M. E. M. Toorians, Conceptualization, Data curation, Investigation, Methodology, Project administration, Writing – review and editing | T. J. Davies, Funding acquisition, Project administration, Resources, Supervision, Writing – review and editing | R. D. Stewart, Funding acquisition, Project administration, Resources, Supervision, Writing – review and editing | W. J. Goosen, Funding acquisition, Project administration, Resources, Supervision, Writing – review and editing | M. A. Miller, Funding acquisition, Project administration, Resources, Supervision, Writing – review and editing

## DATA AVAILABILITY

Sanger *hsp65* sequences of uncultured MTBC from this study are available on NCBI (PX242100, PX242101, and PX242102). Raw *Mycobacterium* sequences are available as supplemental material and labeled by sample number as per Table 2.

## ETHICS APPROVAL

Ethical approval for this study was granted by the Stellenbosch University Biological and Environmental Safety Research Ethics Committee (SU-BES-2023-29171). Section 20 research approval was granted by the South African Department of Agriculture, Land Reform and Rural Development (12/11/1/1/26 [2316SS] and 12/11/1/7/6 [2703KL]). A research agreement (SS705) was approved by South African National Parks for acquiring samples.

## ADDITIONAL FILES

The following material is available online.

### Supplemental Material

**Data S1 (Spectrum01658-25-s0001.txt).** Raw fastq of Mycobacterial hsp65 Sanger sequences. Soil DNA was amplified with a genus specific hsp65 PCR and Sanger sequenced producing a forward and reverse fastq per sample. Consensus sequences which matched a known *Mycobacterium* on the NCBI database after trimming and alignment are included in this list.
**Supplemental tables (Spectrum01658-25-s0002.docx).** Tables S1 and S2.

### Open Peer Review

**PEER REVIEW HISTORY (review-history.pdf).** An accounting of the reviewer comments and feedback.

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
