## [Reviewer comments · Microbiology Spectrum]

Microbiology Spectrum

A microbial safari: finding evidence of *Mycobacterium bovis* DNA in soil from the Kruger National Park, South Africa.

Megan Matthews, Marjolein Toorians, Jonathan Davies, Ross Stewart, Wynand Goosen, and Michelle Miller

Corresponding Author(s): Megan Matthews, Stellenbosch University Faculty of Medicine and Health Sciences

Review Timeline:

Submission Date:	May 29, 2025
Editorial Decision:	July 7, 2025
Revision Received:	September 5, 2025
Accepted:	September 8, 2025

Editor: Artem Rogovsky

Reviewer(s): Disclosure of reviewer identity is with reference to reviewer comments included in decision letter(s). The following individuals involved in review of your submission have agreed to reveal their identity: Jayne Hope (Reviewer #1); Eman Khalifa (Reviewer #2)

Transaction Report:

DOI: <https://doi.org/10.1128/spectrum.01658-25>

Re: Spectrum01658-25 (A microbial safari: finding evidence of *Mycobacterium bovis* DNA in soil from the Kruger National Park, South Africa.)

Dear Dr. Megan Ceris Matthews:

Thank you for the privilege of reviewing your work. Below you will find my comments, instructions from the Spectrum editorial office, and the reviewer comments.

Revision Guidelines

Sincerely,
Artem Rogovsky
Editor
Microbiology Spectrum

Reviewer #1 (Comments for the Author):

The paper demonstrates the presence of *M. bovis* in environmental samples from Kruger National Park where infection with this pathogen is a concern for wildlife and conservation. The findings are clear, and the limitations of the study well presented. Minor comment - the abstract could be improved by adding more detail to indicate that a large number of samples contain *Mycobacteria*, but these are predominantly NTM. This would be of interest and also enhances confidence in the ability to isolate

quality DNA from soil.

Reviewer #2 (Public repository details (Required)):

Sequenced data should be deposited into gene bank and please add the ID number of the DNA submission bank.

Reviewer #2 (Comments for the Author):

Thanks for your kind work and please follow the attached file
Regards

The manuscript submitted to “**Microbiology Spectrum**” entitled: “**A microbial safari: finding evidence of *Mycobacterium bovis* DNA in soil from the Kruger National Park, South Africa**” which investigate the presence of MTBC, or specifically *M. bovis* DNA, in KNP soil suggests potential contamination of the environment. Please correct the following:

- Sequenced data should be deposited into gene bank and please add the ID number of the DNA submission bank.
- You have to collect samples from the animals in the park as you collected samples from the environment in order to study the source of the infection.
- All names of bacteria genus and species should be italic as well as the word *Mycobacterium* or *Mycobacteria* should be italic as *Mycobacterium* or *Mycobacteria* in whole the manuscript even in reference section.
- The presence of *M. bovis* DNA in the environment is not indicative to the presence of the bacteria it self a live, since the DNA can persist for decades, so here you should declare that may or may not found the live bacteria in the environment.
- Also, the presence of *M. bovis* DNA in the environment should be highlighted to the source and the epidemiology of the infection so here you should identify for example by the sequencing analysis to the genetic relatedness of the examined DNA and write the obtained result as a conclusion of your manuscript.
- So, line no. 38 you should remove this sentence (**bacterial shedding from infected animals**) until you found the answer to the previously mentioned 2 points above. The same for this sentence (**that *M. bovis* can be shed by infected animals into water, soil or plant material and could infect animals in the surrounding environment**) at line 49-50.
- Line 52-54, please what is the meaning of this sentence (**The DNA-based detection techniques used in this study could be valuable for detecting other Mycobacterial pathogens of clinical or veterinary importance using non-invasive environmental sampling**)?
- Please remove this sentence (**culture independent detection**) from line 56, as here you did not use this technique.
- Add tab at the start of each paragraph in whole manuscript.
- Introduction is good and organized.
- Please add the missed references in step by step of **Materials and Methods**.
- Result is confused and should be rewritten in more clear form.
- Discussion is too short and shallow, so please write again in more details and discuss with more relevant references.
- Please add a paragraph of conclusion (without citing references), before the section of **Acknowledgements**.

- References:
 - References are updated.
 - More relevant updated references related to the environment of specially parks should be added.
- Please add the authors contributions.
- Figures as well as tables Legends should be summarized and take the style of legends not as paragraphs.
- Kindly improve the font and resolution of all figures.

04 September 2025

Dear Reviewers of Spectrum 01658-25 Manuscript

On behalf of all the authors, we thank you for your helpful and insightful comments and have made changes to the manuscript to address your concerns. We believe the manuscript is now more suitable for publication in the American Society for Microbiology Spectrum Journal. The response to each concern is stipulated below per reviewer.

Reviewer #1 Comments:

The paper demonstrates the presence of *M. bovis* in environmental samples from Kruger National Park where infection with this pathogen is a concern for wildlife and conservation. The findings are clear, and the limitations of the study well presented. Minor comment - the abstract could be improved by adding more detail to indicate that a large number of samples contain *Mycobacteria*, but these are predominantly NTM. This would be of interest and also enhances confidence in the ability to isolate quality DNA from soil.

Response:

We have added the suggested details: “Sanger sequencing detected organisms belonging to *Mycobacterium* genus in 44 samples and non-tuberculous *Mycobacterium* species in 21 samples” to the abstract in lines 34-36 of the ‘clean manuscript’. The wording in lines 28-41 also had to be adjusted to keep the abstract below the 250-word limit.

Reviewer #2 Comments:

The manuscript submitted to “Microbiology Spectrum” entitled: “A microbial safari: finding evidence of *Mycobacterium bovis* DNA in soil from the Kruger National Park, South Africa” which investigate the presence of MTBC, or specifically *M. bovis* DNA, in KNP soil suggests potential contamination of the environment. Please correct the following:

Comment 1:

Sequenced data should be deposited into gene bank and please add the ID number of the DNA submission bank.

Response 1:

Sequenced data for the MTBC positive samples are deposited into NCBI ((PX242100, PX242101 and PX242102) as indicated in lines 353-356 of the clean manuscript. Samples are labelled by number as per the written manuscript (See Table 2). Additionally, the raw fastq for all *Mycobacterium hsp65* Sanger sequences are now available as supplementary material.

Comment 2:

You have to collect samples from the animals in the park as you collected samples from the environment in order to study the source of the infection.

Response 2:

Collecting samples from animals in the park did not fall within the scope of this project, however, these authors and others have previously published data (Bernitz *et al.*, 2021; Lakin *et al.*, 2022; Dwyver *et al.*, 2022; Roos *et al.*, 2023; De Klerk-Lorist *et al.*, 2024a) which demonstrates that multiple wildlife species in KNP are infected with *M. bovis*. Our study focused on determining if there was evidence of MTBC in the KNP environment, which could be a possible risk for TB transmission. This would be the first step in trying to link specific *M. bovis* strains in the environment with those found in infected wildlife. In lines 326-330 of the clean manuscript, we state that further research is required to establish an epidemiological link between soilborne *M. bovis* and the animal population. In lines 330-337 we have added a paragraph describing how this could potentially be investigated in future studies. We hope this clarifies our intention with this study and are open to suggestions on how to make this more apparent in the manuscript.

Comment 3:

All names of bacteria genus and species should be italic as well as the word Mycobacterium or Mycobacteria should be italic as *Mycobacterium* or *Mycobacteria* in whole the manuscript even in reference section.

Response 3:

Bacteria genus and species names have all been placed in italics. We did not italicise 'mycobacterial' as it is generally used without italics to describe culturing or detection techniques (Dawson, 2000; <https://www.cdc.gov/tb/glossary/index.html>).

Comment 4:

The presence of *M. bovis* DNA in the environment is not indicative to the presence of the bacteria itself alive, since the DNA can persist for decades, so here you should declare that may or may not find the live bacteria in the environment.

Response 4:

Thank you for your comment, we agree we could not determine bacterial viability based on DNA evidence. We therefore have stated that we could not confirm viability (lines 39-41 and 317-325 of the clean manuscript) and this is now reemphasized in line 349-352 as part of the requested conclusion statement.

Comment 5:

The presence of *M. bovis* DNA in the environment should be highlighted to the source and the epidemiology of the infection so here you should identify for example by the sequencing analysis to the genetic relatedness of the examined DNA and write the obtained result as a conclusion of your manuscript.

Response 5:

One of the stated limitations was that we were unable to perform mycobacterial culture to provide material for WGS, which could then be compared to sequences of *M. bovis* isolates from KNP wildlife. The *hsp65* gene sequences we obtained in this study are conserved between members of MTBC. The RD amplicons provide differentiation by MTBC species but would not provide sufficient data to examine the phylogenetic relatedness of environmental and animal isolates. For example, the Sanger sequence of our positive control (confirmed *M. bovis* isolate) had 100% coverage and identity with a *Mycobacterium tuberculosis* 1821ADB41 sequence on NCBI. The *hsp65* and RD regions are therefore not recommended for evaluating diversity among *M. bovis* isolates and any variation observed would most likely be due to poor sequence quality or contaminants in the environmental samples. However, we acknowledge that NGS/WGS of both environmental and animal isolates will be the next step to explore the source of soil contamination by MTBC. We have better addressed this limitation in the discussion (lines 326-337 of the clean manuscript).

Comment 6:

So, line no. 38 you should remove this sentence (bacterial shedding from infected animals) until you found the answer to the previously mentioned 2 points above. The same for this sentence (that *M. bovis* can be shed by infected animals into water, soil or plant material and could infect animals in the surrounding environment) at line 49-50.

Response 6:

Thank you for your comment. We agree the statements in line 38 (line 37-39 in the clean manuscript) are not proven but are possible based on scientific literature which indicates that *M. bovis* is endemic in KNP wildlife populations (Davey, 2023), the pathogen can be shed by infected wildlife in the KNP (Meiring *et al.*, 2021; Dwyer *et al.*, 2024) and bacteria shed can persist in the soil environment if conditions are favourable (Fine *et al.*, 2011). Additionally,

as presented in the introduction (lines 78-85 of the clean manuscript), experimental studies have shown that environmental *M. bovis* can cause infection in animals (Phillips *et al.*, 2003; Palmer *et al.*, 2004). While the findings from the current study did not attempt to identify the animal source of *M. bovis* in our environmental samples, shedding associated with the high prevalence of TB in African buffalo and other wildlife species in KNP appears to be the most likely explanation.

The statement in lines 49-50 (lines 49-51 in the clean manuscript) refers to conclusions made by other studies in *M. bovis* endemic areas in Europe (e.g. Pereira *et al.*, 2024) where there were some similarities to the multiple-host ecosystem and molecular methods used in our study. We would therefore like to keep these statements in the manuscript but are open to suggestions on how they could be re-worded to better communicate that they are theoretical rather than definitive.

Comment 7:

Line 52-54, please what is the meaning of this sentence (The DNA-based detection techniques used in this study could be valuable for detecting other Mycobacterial pathogens of clinical or veterinary importance using non-invasive environmental sampling)?

Response 7:

Non-invasive environmental sampling for clinical or veterinary fields mentioned in line 53-56 of the clean manuscript refers to taking soil, water or vegetation from an area rather than invasive samples (e.g. tissue/respiratory secretions) directly from animal or human hosts to investigate the presence of pathogens. For example, wastewater sampling was used for non-invasive/non-intrusive environmental sampling of SARS-CoV-19 strains from the human population during the pandemic (Di Maria *et al.*, 2021; Wong *et al.*, 2021). This has been expanded in lines 55-56 of the clean manuscript.

Comment 8:

Please remove this sentence (culture independent detection) from line 56, as here you did not use this technique.

Response 8:

A culture independent technique is when you do not culture *Mycobacterium* prior to detection with molecular tests, e.g. DNA extraction and PCR. We did not culture soil samples, but we did culture a known *M. bovis* isolate to serve as a positive control (see lines 129-137 of the clean manuscript) in this study. Culturing of soil samples was not possible due to controlled disease (FMD) biosecurity restrictions and so we would respectively like to keep the sentence in the manuscript.

Comment 9:

Add tab at the start of each paragraph in whole manuscript.

Response 9:

The journal does not require tabs be used to separate paragraphs; however, we have included them in the introduction, materials and methods and discussion sections for the reviewer. The other sections have paragraphs separated by headings. We will follow the advice of the editor as to whether the tabs are included in the published version of the manuscript

Comment 10:

Introduction is good and organized.

Response 10:

Thank you for your comment.

Comment 11:

Please add the missed references in step by step of Materials and Methods.

Response 11:

We believe references were included where appropriate in the methodology. References did occasionally refer to multiple steps, but this was necessary to provide sufficient detail from available literature to repeat the described experiments successfully. If the reviewer believes we are missing citations we ask for further clarity as to where we can add them.

Comment 12:

Result is confused and should be rewritten in more clear form.

Response 12:

We apologise if you found the results section confusing. It was organised to follow the methods section. We have revised the headings to make it clear to the reviewer but are open to any additional suggestions.

Comment 13:

Discussion is too short and shallow, so please write again in more details and discuss with more relevant references.

Response 13:

Our aim was to present a succinct and concise Discussion of our results. There is a paucity of literature on environmental *M. bovis*, particularly in Africa, which we hope we have cited appropriately. We have added further references and a paragraph to the discussion (lines 326-337 and line 283 of the clean manuscript) to explain study limitations and context more fully.

If you think the Discussion should be expanded to include additional content, we would be grateful if you could indicate which topics you would consider to be most relevant.

Comment 14:

Please add a paragraph of conclusion (without citing references), before the section of Acknowledgements.

Response 14:

We have now added a conclusion section in lines 346-352 of the clean manuscript.

Comment 15:

More relevant updated references related to the environment of specially parks should be added.

Response 15:

References related to environmental *M. bovis* in wildlife parks were added in lines 81-84, line 265 and lines 292-296 of the clean manuscript. Please advise if there are additional references you would like included.

Comment 16:

Please add the authors contributions.

Response 16:

Author contributions are not added in the body of the manuscript in this Journal. Rather the information is entered when submitting a publication and made available to readers of the published manuscript when they click author info and affiliations on top of the page (see below figure).

Comment 17:

Figures as well as tables Legends should be summarized and take the style of legends not as paragraphs.

Response 17:

Figure and table legends have been summarized in lines 610-617, and 626-679 of the clean manuscript.

Comment 18:

Kindly improve the font and resolution of all figures.

Response 17:

The font and resolution of all images has been improved and tiff files compressed to fit the 100 mb submission upload limit.

References

- Bernitz N, Kerr TJ, Goosen WJ, Chileshe J, Higgitt RL, Roos EO, Meiring C, Gumbo R, de Waal C, Clarke C, Smith K, Goldswain S, Sylvester TT, Kleynhans L, Dippenaar A, Buss PE, Cooper DV, Lyashchenko KP, Warren RM, van Helden PD, Parsons SDC, Miller MA. 2021. Review of diagnostic tests for detection of *Mycobacterium bovis* infection in South African wildlife. *Front Vet Sci* 8:588697. doi:10.3389/fvets.2021.588697
- Davey, S., 2023. Challenges to the control of *Mycobacterium bovis* in livestock and wildlife populations in the South African context. *Irish Veterinary Journal*, 76(Suppl 1), p.14. <https://doi.org/10.1186/s13620-023-00246-9>.
- Dawson, D.J., 2000. Mycobacterial terminology. *Journal of Clinical Microbiology*, 38(10), pp.3913-3913.
- De Klerk-Lorist, L.M., Miller, M.A., Mitchell, E.P., Lorist, R., Van Dyk, D.S., Mathebula, N., Goosen, L., Dwyer-Leonard, R., Ghielmetti, G., Streicher, E.M. and Kerr, T.J., 2024. Case report: Discovery of tuberculosis caused by *Mycobacterium bovis* in free-ranging vervet monkeys in the Greater Kruger Conservation Area. *Frontiers in Veterinary Science*, 11, p.1460115.
- Di Maria, F., La Rosa, G., Bonato, T., Pivato, A., Piazza, R., Mancini, P., Ferraro, G.B., Veneri, C., Iaconelli, M., Beccaloni, E. and Scaini, F., 2021. An innovative approach for the non-invasive surveillance of communities and early detection of SARS-CoV-2 via solid waste analysis. *Science of The Total Environment*, 801, p.149743.
- Dwyer R, Goosen W, Buss P, Kedward S, Manmela T, Hausler G, Chileshe J, Rossouw L, Fowler JH, Miller M, Witte C. 2022. Epidemiology of *Mycobacterium bovis* infection in free-ranging rhinoceros in Kruger National Park, South Africa. *Proc Natl Acad Sci* 119(24):e2120656119. doi.org/10.1073/pnas.2120656119

- Dwyer, R., Witte, C., Buss, P., Warren, R., Miller, M. and Goosen, W., 2024. Antemortem detection of *Mycobacterium bovis* in nasal swabs from African rhinoceros. *Scientific Reports*, 14(1), p.357.
- Fine, A.E., Bolin, C.A., Gardiner, J.C. and Kaneene, J.B., 2011. A study of the persistence of *Mycobacterium bovis* in the environment under natural weather conditions in Michigan, USA. *Veterinary Medicine International*, 2011(1), p.765430. <https://doi.org/10.4061/2011/765430>.
- Lakin HA, Tavalire H, Sakamoto K, Buss P, Miller M, Budischak SA, Raum K, Ezenwa VO, Beechler B, Jolles A. 2022. Bovine tuberculosis in African buffalo (*Syncerus caffer*): progression of pathology during infection. *PLoS Negl Trop Dis* 16(11):e0010906. doi.org/10.1371/journal.pntd.0010906
- Meiring, C., Higgitt, R., Goosen, W.J., van Schalkwyk, L., de Klerk-Lorist, L.M., Buss, P., van Helden, P.D., Parsons, S.D., Möller, M. and Miller, M., 2021. Shedding of *Mycobacterium bovis* in respiratory secretions of free-ranging wild dogs (*Lycaon pictus*): implications for intraspecies transmission. *Transboundary and Emerging Diseases*, 68(4), pp.2581-2588.
- Palmer M. V., Waters W. R., and Whipple D. L., Shared feed as a means of deer-to-deer transmission of *Mycobacterium bovis*, *Journal of Wildlife Diseases*. (2004) **40**, no. 1, 87–91, 2-s2.0-3543081563.
- Phillips, C.J.C., Foster, C.R.W., Morris, P.A. and Teverson, R., 2003. The transmission of *Mycobacterium bovis* infection to cattle. *Research in veterinary science*, 74(1), pp.1-15.
- Pereira, A.C., Pinto, D. and Cunha, M.V., 2024. First time whole genome sequencing of *Mycobacterium bovis* from the environment supports transmission at the animal-environment interface. *Journal of Hazardous Materials*, 472, p.134473. <https://doi.org/10.1016/j.jhazmat.2024.134473>.
- Roos, E.O., Loubser, J., Kerr, T.J., Dippenaar, A., Streicher, E., Olea-Popelka, F., Robbe-Austerman, S., Stuber, T., Buss, P., de Klerk-Lorist, L.M. and Warren, R.M., 2023. Whole genome sequencing improves the discrimination between *Mycobacterium bovis* strains on the southern border of Kruger National Park, South Africa. *One Health*, 17, p.100654.
- Wong, J.C.C., Tan, J., Lim, Y.X., Arivalan, S., Hapuarachchi, H.C., Mailepessov, D., Griffiths, J., Jayarajah, P., Setoh, Y.X., Tien, W.P. and Low, S.L., 2021. Non-intrusive wastewater surveillance for monitoring of a residential building for COVID-19 cases. *Science of The Total Environment*, 786, p.147419.

Re: Spectrum01658-25R1 (A microbial safari: finding evidence of *Mycobacterium bovis* DNA in soil from the Kruger National Park, South Africa.)

Dear Dr. Megan C Matthews:

Your manuscript has been accepted, and I am forwarding it to the ASM production staff for publication. Your paper will first be checked to make sure all elements meet the technical requirements. ASM staff will contact you if anything needs to be revised before copyediting and production can begin. Otherwise, you will be notified when your proofs are ready to be viewed.

Sincerely,
Artem Rogovsky
Editor
Microbiology Spectrum